# Follicular Lymphoma in the 5th Edition of the WHO-Classification of Haematolymphoid Neoplasms—Updated Classification and New Biological Data

**DOI:** 10.3390/cancers15030785

**Published:** 2023-01-27

**Authors:** Katrin S. Kurz, Sabrina Kalmbach, Michaela Ott, Annette M. Staiger, German Ott, Heike Horn

**Affiliations:** 1Department of Clinical Pathology, Robert-Bosch-Krankenhaus, 70376 Stuttgart, Germany; 2Dr. Margarete Fischer-Bosch-Institute of Clinical Pharmacology, University of Tübingen, 70376 Stuttgart, Germany; 3Department of Pathology, Marienhospital, 70199 Stuttgart, Germany

**Keywords:** follicular lymphoma (FL), classification, classic FL (cFL), WHO-HAEM5, localized stage FL (lFL), systemic FL (sFL), *BCL2* translocation

## Abstract

**Simple Summary:**

In this paper, we give an overview on the classification of Follicular lymphoma as modified in the 5th edition of the WHO classification of haematolymphoid tumors and discuss new data regarding the biological background of the classification.

**Abstract:**

The conceptual description of Follicular lymphoma (FL) in the 5th edition of the World Health Organization (WHO) classification of haematolymphoid tumors (WHO-HAEM5) has undergone significant revision. The vast majority of FL (85%) with a follicular growth pattern are composed of centrocytes and centroblasts, harbor the t(14;18)(q32;q21) translocation and are now termed classic FL (cFL). They are set apart from three related subtypes, FL with predominantly follicular growth pattern, FL with unusual cytological features (uFL) and follicular large B-cell lymphoma (FLBCL). In contrast to the revised 4th edition of the WHO classification of haematolymphoid tumors (WHO-HAEM4R), grading of cFL is no longer mandatory. FL with a predominantly diffuse growth pattern had been previously recognized in WHO-HAEM4R. It frequently occurs as a large tumor in the inguinal region and is associated with CD23 expression. An absence of the *IGH::BCL2* fusion and frequent *STAT6* mutations along with 1p36 deletion or *TNFRSF14* mutation is typical. The newly introduced subtype of uFL includes two subsets that significantly diverge from cFL: one with “blastoid” and one with “large centrocyte” variant cytological features. uFL more frequently displays variant immunophenotypic and genotypic features. FLBCL is largely identical to WHO-HAEM4R FL grade 3B and renaming was done for reasons of consistency throughout the classification. In-situ follicular B-cell neoplasm, pediatric-type FL, duodenal-type FL and primary cutaneous follicle center lymphoma are categorized as discrete entities. In addition, novel findings concerning underlying biological mechanisms in the pathogenesis of early and systemic follicular lymphoma will be presented.

## 1. Introduction

Since the 3rd edition of the WHO classification of lymphoid tumors has appeared in 2001 [1], it has constituted the worldwide reference laying down the basic principles for the diagnosis of lymphoid tumors. Successive WHO volumes that were published in 2008 [2] and 2017 [3] have broadened and refined existing knowledge on this still challenging type of neoplasm. Now, the upcoming 5th edition of the WHO classification of haematolymphoid tumors (WHO-HAEM5), due to appear in 2023, has updated the conceptual framework and the major developments that have occurred in the field since 2017. Naturally, WHO-HAEM5 constitutes a systematic evolution of the prior classifications. WHO-HAEM5 has introduced—as have the other volumes of the 5th edition—a hierarchical system of classification in organizing diseases in order of increasing levels of specification into category, family/class, entity/type and subtype. At the same time, WHO-HAEM5 has recognized the increasing importance of genetic data in the definition and risk stratification of lymphoid neoplasia and has incorporated these data to the extent meaningful and acknowledged in the scientific community. In addition, for the first time, non-neoplastic conditions possibly representing a caveat in the differential diagnosis mimicking lymphoma have been introduced.

In this work we delineate the major changes that have occurred in the chapters on follicular lymphomas (FL) that in part have undergone significant revision. New genetic and molecular data on FL and their impact on the classification and definition of these diseases are described.

FL represents, next to diffuse large B-cell-lymphoma (DLBCL), the most common B-cell neoplasm in the Western world [4]. According to both WHO-HAEM5 and the proposal of the International Consensus Classification (ICC) [5], it is defined as a tumor of germinal center B-cells composed of varying proportions of centrocytes and centroblasts with an at least partially follicular growth pattern. Rare cases, however, can present with an entirely diffuse growth, and mixed follicular and diffuse growth patterns are frequently seen. FL is mainly a nodal disease, but involvement of the spleen and bone marrow is frequently encountered. However, in systemic disease, both lymphatic tissues such as Waldeyer ring in the tonsil or Peyer patches in the gastrointestinal tract as well as extra lymphatic tissues such as the ocular adnexa, soft tissue, liver and skin can be affected. FL primarily arising in extra lymphatic tissues, on the other hand, is rare, but has been reported in the skin, the testis, the ovary and other organs or organ systems.

FL is an indolent disease and overall, the clinical course is characterized by multiple relapses, and can be surprisingly variable [6,7]. Most patients are diagnosed with systemic disease including widespread central and peripheral lymph node involvement, splenomegaly and bone marrow infiltration. However, nodal disease can also present in localized clinical stages I or II [6]. The clinical course is protracted, and patients may be asymptomatic without need for treatment at diagnosis or present with advanced-stage disease and a high tumor burden. 

Therapy usually involves rituximab combined with single or multi-agent therapy, for example with CHOP (Cyclophosphamide, Doxorubicin, Vincristine, Prednisone) or Bendamustine. In general, the prognosis is inferior in patients with relapse occurring within 2 years of therapy and in those whose tumors transform to aggressive lymphoma, most often DLBCL. Both clinical parameters such as the follicular lymphoma international prognostic index (FLIPI) [8] and combined clinico-molecular risk factors such as the m7-FLIPI [9] can predict survival in patients with FL. Transformation of FL to DLBCL usually heralds an inferior prognosis and occurs at a risk of 2% per year [10].

## 2. WHO-HAEM5 and the Chapter on FL

The vast majority of FL show an at least partially follicular growth pattern and are composed of centrocytes and centroblasts. The t(14;18) is regarded as the initiating event in these tumors. In contrast, FL with a varying growth pattern or variant cytologies are much rarer, as are the more specific entities of FL in WHO-HAEM5 such as in-situ follicular B-cell neoplasm, duodenal-type FL, pediatric-type FL, and primary cutaneous follicle center lymphoma (Table 1). Therefore, the WHO-HAEM5 proposed an entirely new concept to subclassify FL. Basing on their frequency, FL with a (partially) follicular growth pattern, composed of centrocytes and centroblasts and harboring the t(14;18) in around 85% of cases are now termed *classic FL* (cFL) and are set apart from the rarer subtypes. In essence, FL grades 1, 2 and 3A of WHO-HAEM4R will now fall into this entity.

At the same time, WHO-HAEM5 takes into account the results of various studies that have reported the poor reproducibility of grading which is usually done by counting the number of centroblasts/transformed cells in 10 consecutive high power fields [11,12]. Apart from technical issues related to inadequate sampling, the definition and the recognition of centroblasts is an issue in the poor reproducibility of this grading system. In addition, the majority of clinical trials using modern treatment protocols have failed to provide evidence of significant differences in the clinical outcome between FL grades 1/2 and 3A [11,13,14], and many clinicians tend to treat patients with cFL with similar protocols. Therefore, WHO-HAEM5 states that the grading of cFL should be optional, while the ICC proposal still recommends grading of FL [5].

Rare cases of FL, the vast majority of them with low blast count, present with a predominantly diffuse growth pattern, and these are set apart from, yet regarded as related to, cFL because of their similar cytological features. Two other FL variants with uncommon cytological features preclude grading in the established system. These cases mainly have a follicular growth pattern and consist of either medium-sized “blastoid“ or “large centrocytic“ cells, and are regarded as subtypes with unusual cytological features. The latter three uncommon FL subtypes do also show a different genetic make-up in comparison with cFL; they are often t(14;18) negative and may also fail to express the BCL2 protein. These features also represent a rationale to separate them from cFL, although their genetic landscape is similar including mutations in genes typically encountered also in cFL such as *CREBBP*, *STAT6*, *TNFRSF14*, and others.

## 3. In-Situ Follicular B-Cell Neoplasm

The new term of In-situ follicular B-cell-neoplasm (ISFN) in WHO-HAEM5 is equivalent to in-situ follicular neoplasia in WHO-HAEM4R and the ICC proposal [5]. ISFN is defined as a partial or complete colonization of rare or some reactive germinal centers (GCs) by follicular (CD10+) B-cells that carry the *IGH::BCL2* fusion and display strong expression of BCL2. It is rare, as a rule fortuitous, finding in lymph nodes or lymphatic tissues that have been removed for other reasons. In larger series, it has been described in 2% of lymph nodes [15,16]. In the vast majority of cases, no subsequent FL will develop; however, in some patients, staging procedures reveal concurrent established FL or other types of lymphoma such as mantle cell lymphoma or chronic lymphocytic leukemia [15,17]. Only a small number of patients will ever develop FL or other types of lymphoma subsequent to the diagnosis of ISFN.

Histologically, the condition may be suspected owing to the monotonous appearance of a few or a number of GCs that are mainly harboring centrocytes. However, these changes are easily overlooked and ISFN is usually recognized only upon BCL2 staining revealing the partial or complete colonization of GCs by strongly BCL2 positive B cells (Figure 1) [18]. As a rule, CD10 expression—in contrast with reactive follicles—is also unusually strong.

Conceptually, ISFN maybe be viewed as the first recognizable tissue manifestation of circulating t(14;18)-positive FL-like B-cells that already have acquired additional mutations. The condition is invariably t(14;18) positive and clonal relationship has been shown between circulating *IGH::BCL2* fusion-positive cells and ISFN in the same patient [19,20]. In addition, mutations in *CREBBP*, *KMT2D* und *TNFRSF14*, deletions in 1p36, and acquisition of N-glycosylation sites have been reported [21,22,23,24,25,26]. Cases of FL that are preceded by an ISFN have acquired additional mutations to those already present in the ISFN such as in *EZH2* [21,27,28,29]. Up to now, no t(14;18)-negative ISFN has been reported.

## 4. Classic FL

The histopathological changes seen in follicular FL mimic those seen in follicular hyperplasia. In some cases, in spite of manifest FL infiltration in the lymph node, the architecture may appear only slightly altered. In cFL, there is an entirely or partially follicular growth pattern with uniformly appearing, numerous, closely packed follicles that may slightly vary in size and shape. Diffuse areas may be present to a variable extent, and there may be spread beyond the lymph node capsule and/or sclerosis. The neoplastic GCs are composed of centrocytes and centroblasts and in the large majority of cases, centrocytes—small to medium-sized cells with scant cytoplasm and slightly angulated nuclei—predominate (Figure 2A,B); WHO-HAEM4R continued to recommend grading of FL basing on the number of centroblasts/large transformed cells in 10 random neoplastic follicles at high magnification (×400) and that correction for the different sizes of the microscopic fields be made. As mentioned, WHO-HAEM5 considers grading as optional but if done, recommends grading of FL according to the number of centroblasts per mm (<90 or >90 per mm^2^ for a HPF size of 0.159 mm diameter and 0.458 mm^2^) [30]. Characteristically, the starry-sky pattern—the usual bipartition of reactive GCs into a “dark” zone and a “light” zone is absent. The interfollicular areas are involved by lymphoma cells to a various extent and upon CD20 staining, there may be a prominent interfollicular (at times also CD10 positive) tumor cell component which forms an integral part of the tumor.

The immunophenotype of cFL is uniform. The neoplastic cells express pan B-cell antigens such as CD19, CD20 and PAX5 and, as a rule, are positive for germinal center-associated markers such as CD10, BCL6, HGAL, and others [31,32]; within the neoplastic follicles, the number of infiltrating T-cells is variable and ranges from few to abundant. In 80–90% of the cases, the neoplastic follicles also express the BCL2 protein so that in a follicular proliferation, the simultaneous expression of CD10 and BCL2 in the GCs is a clear hint to the neoplastic nature of the process. The BCL2 expression in the neoplastic follicles, in most cases, is due to the t(14;18)(q32;q21)/*IGH::BCL2* fusion which leads to constitutive (over-)expression of the protein. In cases where the neoplastic cells do not carry the t(14;18) translocation, the neoplastic GCs may be either BCL2 protein negative or only weakly and inconsistently positive; in some cases that are BCL2 protein negative in spite of the presence of the translocation, mutations in the *BCL2* gene can alter the binding site of the antibody, requiring testing of additional antibodies directed against other epitopes [33]. The Ki67 proliferation index in cFL usually is distinctly lower in neoplastic follicles than in reactive lympho-follicular hyperplasia [34].

The t(14;18) is commonly regarded as the initiating event in 85–90% cFL.

The founder cell of cFL is a pre-B cell in which an error occurs at VDJ recombination leading to the t(14;18). Of importance, circulating t(14;18)-positive B-cells can be found in the peripheral blood at very low levels in >70% of healthy adults. These cells represent memory B-cells that have passed the GC. The number of these recirculating FL-like B cells in the peripheral blood increases with age. Environmental exposure for example to pesticides seems to lead to higher levels [35]. Via repeated re-entries into the GC, the risk of acquisition of additional genetic alterations increases [20]. These B-cells are somatically hypermutated and V-regions have acquired N-glycosylation sites. The vast majority of carriers of t(14;18)-positive B-cells will never develop overt lymphoma [36,37]; however, ISFN can be observed as a fortuitous event in lymph nodes removed for other reasons, and this particular condition obviously represents the first morphologically recognizable manifestation of these cells in tissues (see above). Of note, at this stage, additional genetic events such as for example mutations in *CREBBP* have already taken place. In most FL, branching evolution can be traced by molecular analysis, leading to multiple subclones already at early disease [38,39]. Genetic events associated with the establishment and progression of the malignant clone are chromosomal gains of 1q, 2p, 7, 6p, 8q, 17q, 18 and X, and deletions are recurrently observed in 1p, 6q, 9p and 10q [40]. The most frequent mutations target *KMT2D* and *CREBBP, BCL2, EZH2, ARID1A, MEF2B* and *KMT2C* [38,39,41] Deletion in 1p36 leads to loss of heterozygosity (LOH) of *TNFRSF14* that can also be altered by mutations [42,43]. More uncommon gene mutations are found in *STAT6*, *CARD11*, *FOXO1* and others. Figure 3 gives an overview of the scenario of FL development and progression.

Translocations involving *BCL6* are encountered in cFL in concert with those of *BCL2* or without the t(14;18). They have been described in 10 to 20% of t(14;18)-positive and -negative FL [44,45,46,47].

About 10–15% of morphologically typical cFL are negative for the *IGH*::*BCL2* fusion, and recent work has delineated some differences to t(14;18)-positive FL. Their gene expression (GE) and microRNA profile resembles that of an exit GC cell population [48,49]. Although the mutational and the SNV spectrum of *BCL2*-negative FL is similar to that of typical (*BCL2*-positive) cFL, there are some differences with more frequent mutations in *STAT6* and a lower rate of *KMT2D* mutations [50,51]. *KMT2D* deficiency associated with loss-of-function mutations has been reported to possibly drive GC expansion due to enhanced proliferation of lymphocytes and to impede B cell differentiation [52]. The lower frequency of *KMT2D* mutations in *BCL2*-negative FL fits well into the concept of a late GC phenotype in these lymphomas [47,48]. Moreover, *BCL2*-negative FL have shown a prominent or exclusive enrichment of single nucleotide variants (SNVs) associated with the microenvironment and N-glycosylation, while newly acquired N-glycosylation sites (NANGS) are reduced compared to *BCL2*-positive FL [50,53]. NANGS are usually generated in the IGVH regions and are required for lectin-mediated interaction with non-malignant bystander cells (e.g., follicular dendritic cells and macrophages), again emphasizing distinct underlying mechanisms driving the pathogenesis of *BCL2* translocation positive and -negative FL.

At diagnosis, the majority of cFL presents with symptomatic disease in advanced/systemic stages of FL (sFL). In contrast only around 10 to 15% of cFL are diagnosed in localized stages (lFL). Mechanisms underlying the different clinical appearances have been the focus of FL research within the past decade, and advances in genetic analysis shed light on the biological processes driving FL pathogenesis and progression. However, biological knowledge is predominantly based on data derived from global sFL analysis and only small cohorts of lFL were characterized intensively so far. As described above, it is actually well-known that the hallmark *BCL2* translocation is less frequently observed in lFL, although presence of the translocation does not impair progression free survival and overall survival [46,54]. However, in few patients, rapid early progression (within 2 years) was observed in *BCL2* translocation-negative lFL and sFL, not seen in *BCL2* translocation-positive FL [46,54].

Furthermore, a GE classifier was established allowing for i) the discrimination of lFL and sFL and for ii) the identification of lFL more closely resembling sFL with an inferior clinical outcome [55]. These data support the concept that GE signatures are potentially useful to identify patients in need of intensified therapeutic regimens and to avoid over-treatment. It has been recently reported that lFL and sFL have similar mutational profiles and do not differ significantly in their global genetic aberrations. Nevertheless, different clusters were described in lFL stage I predominantly determined by the presence of *BCL2* and *BCL6* translocations, as well as *CREBBP* and *STAT6* mutations [56]. However, exomic sequencing of lFL has not been performed until now, thus still awaiting more in-depth data of lFL. Of special interest, Los-de Vries and colleagues detected *BCL6* translocations in only 6% of lFL stage I analyzed, which is in clear contrast to the frequencies hitherto reported for sFL [56].

Most data on the specific tumor microenvironment of FL have been gathered from cFL as defined according to WHO-HAEM5. As is implied by their specific niche in the lymph node, FL cells depend on the presence of follicular dendritic cells, T-follicular helper cells and tumor-associated macrophages [57,58]. Within the GC, increased numbers of CD4+ T-cells are present and immune signaling pathways are altered enabling FL cells to escape from immune surveillance [58]. The specific composition of the tumor microenvironment obviously also has prognostic importance [57] and consequently has also been reported to be associated with disease stage [59]. The complexity in tumor and non-malignant bystander cell communication and interaction is in the focus of various new projects, essentially fostering an optimized therapeutic intervention [60] and developing novel 3D models to improve preclinical drug testing [61].

## 5. Follicular Lymphoma with Unusual Cytologic Features

WHO-HAEM5 opposes a subgroup of FL with unusual cytological features (uFL) to cFL. In these—admittedly rare—cases, the tumor cells are either medium-sized blastoid cells or large centrocytes (LCC), so that grading according to the established criterion of counting centroblasts cannot be performed [62,63,64]. For these lymphomas the term “follicular lymphoma, unclassifiable” e.g., with blastoid features has been proposed in WHO-HAEM4R. The blastoid variant consists of medium-sized cells with a narrow rim of cytoplasm and round nuclei with finely dispersed chromatin reminiscent of lymphoblasts (Figure 4). In the LCC variant, the tumor cells are large, but rather resemble (large) centrocytes than typical (small) blasts. The significance of these findings is not clear. At least some studies have reported a clinical course that is inferior to cFL [63]. Often, such cases show higher proliferation indices and/or uncommon phenotypes such as negativity for CD10 or (strong) expression of MUM1. In addition, the frequency of *BCL2* rearrangements has been reported to be distinctly lower than in cFL [63,64].

## 6. Predominantly Diffuse FL

Lymphomas with an entirely or predominantly diffuse growth pattern (DFL) are rare. Especially in small biopsies, where the follicular component can easily be missed, the diagnosis must be made with caution. WHO-HAEM5 regards DFL as related to cFL because of their cytological composition, yet distinct because of their varying immunophenotypic and genetic characteristics [51,65,66]. In this distinctive subtype of FL, there is a diffuse infiltration of predominantly centrocytes and some interspersed centroblasts (Figure 5), where the centrocytic cells seem to have rounder nuclear contours in comparison with their cFL counterparts. These tumors have mainly, but not exclusively, been described in the inguinal region and are as a rule diagnosed in clinical stages I or II [65]. In many cases, small follicular structures, so called micro follicles, are present that variably express CD10 and BCL2 and usually have a high proliferation index. In general, the tumor cells are CD23 and BCL2 positive, while CD10 and BCL6 are variably expressed. It should be noted that this immunophenotypic constellation, especially the reactivity for CD23, is not confined to DFL, but can also be seen in diffuse areas of otherwise typical follicular FL.

DFL, upon GE analysis, cluster within cFL and are usually negative for the t(14;18) chromosome translocation [65]. 50–90% of cases have been reported to harbor a recurrent deletion or copy number neutral loss of heterozygosity (CNLOH) involving 1p36, spanning the *TNFRSF14* gene, as well as recurrent losses and/or CNLOH of 16p13 containing *CREBBP*, *CIITA* and *SOCS1* genes [51,66,67,68]. In addition, mutations in *STAT6* and *CREBBP* have been described [66,67,68]. Mutations in the *STAT6* gene predominantly affect the DNA binding domain, necessary for transactivation of target genes. Apart from activation of the JAK/STAT signaling pathway [69], the antiapoptotic *BCL*-*xL* (*BCL2L1*) represents an important target of *STAT6* [70] that might act as a surrogate for BCL2. Of interest, it has been reported that *STAT6* mutations correlate with positive nuclear staining for phospho-STAT6 [66] and with CD23 expression [51,66]. In the ICC proposal, a new provisional entity, *BCL2* rearrangement-negative, CD23-positive follicle center lymphoma is recognized that obviously contains many cases of DFL; however, is not restricted to FL with diffuse growth pattern [5].

## 7. Follicular Large B-Cell Lymphoma

Follicular large B-cell lymphoma (FLBCL) in WHO-HAEM5 is the new name for FL grade 3B according to the definition given in WHO-HAEM4R and retained in the ICC [5]. It consists of GCs that exclusively harbor centroblasts, while—at least typical—centrocytes are absent. The change of the designation of this subtype from FL3B to FLBCL was made to stress the close relationship of FLBCL to DLBCL. Therefore, although FLBCL still is regarded as a subtype of FL, it has conceptually moved closer to DLBCL in WHO-HAEM5.

Some of these centroblasts/large transformed cells may be small to medium-sized, but they lack the characteristic cytological features of centrocytes (Figure 6A,B) [71]. This subtype is very rare among follicular lymphomas and more often, FLBCL is seen in association with a DLBCL component [11,63,72,73]. In contrast with cFL, CD10 and BCL2 are expressed in only 40% and 50% of cases, respectively. Frequent monotypic cytoplasmic IG-expression (plasmacytic differentiation) has been described [71]. *BCL2* translocations have been seen in less than 10% of cases, and also the frequency of translocations involving *BCL6* is lower than that seen in both cFL and DLBCL. Translocations involving *MYC* have been noted in around 20% of cases [64]. Interestingly, the genetic landscape of cases that harbor both a FLBCL and DLBCL component is more similar to typical DLBCL and there is a discussion if the follicular component in these cases merely represents follicular colonization of pre-existing follicles by DLBCL [64]. Analyzing the gene expression profiles of FLBCL with a DLBCL component indicated that it obviously represents a composite form of FLBCL and DLBCL, nevertheless closely resembling DLBCL [74].

## 8. Pediatric-Type Follicular Lymphoma

In WHO-HAEM5, Pediatric-type follicular lymphoma (PTFL) is defined as a localized nodal B-cell lymphoma occurring primarily in pediatric and adolescent patients [75,76]. It shows a purely follicular growth pattern of clonal GC B-cells with an elevated proliferation index and absence of *BCL2*, *BCL6* and *IRF4* rearrangements. Most cases have been diagnosed in head and neck lymph nodes [75,77]. In the great majority of cases, there is painless localized lymph node enlargement without B-symptoms. Altogether, PTFL is rare, accounting for 1–2% of all childhood NHL and affects predominantly male patients (M:F = 10:1) [77,78]. Histologic examination often reveals a rim of preserved architecture at the lymph node periphery thus creating a “node-in-node” appearance. The conspicuous GCs are enlarged, often irregular shaped with plump protrusions or dumbbell-shaped with attenuated mantle zones and contain intermediate-sized blastoid or large cells, often with a preserved “starry sky” pattern (Figure 7). The immunophenotype resembles that of reactive follicular hyperplasia; most cases are positive for CD10, all are positive for BCL6 and negative for BCL2, though weak reactivity may occur. The proliferation index is high, usually >70% and sometimes, polarization of GCs is still preserved. In paraffin-embedded tissues, light chain restriction may rarely be observed.

By definition, PTFL lacks translocations of *BCL2*, *BCL6*, *MYC* and *IRF4* [43,75,76,79,80]. Proof of clonality (by immunophenotyping or genetics) is an essential diagnostic criterion. The landscape of mutations in PTFL differs from that of cFL in that mutations in epigenetic modifier genes such as *EP300*, *CREBPP*, *EZH2*, *KMT2D* or *ARID1A* frequently observed in cFL are rare [76]. In contrast recurrent alterations are seen in 1p36 (*TNFRSF14*) and mutations of the MAPK pathway such as in *MAP2K1* have been described, next to others [43,76,79,81].

Primary FL of the testis, also described in children or adolescents shows some similarities with PTFL such as localized disease, intermediate to large cell cytology, and the immunophenotype.

## 9. Duodenal-Type Follicular Lymphoma

Duodenal-type FL (DTFL) is a particular extranodal entity of FL that mainly involves the second part of the duodenum; however, manifestations of DTFL can be found throughout the small bowel and, more rarely, in the stomach and colorectum [82]. In most of the cases, DTFL is incidentally detected on endoscopy performed for unrelated reasons, and reveals granular lesions or small or larger polyps up to 2 cm. Usually, DTFL is a very indolent lymphoma staying localized for years [83,84]. It has to be differentiated from primary intestinal FL often showing transmural infiltration and systemic spread, and secondary infiltration of the intestines by generalized, sFL.

On microscopic examination, the atypical follicles mainly occupy the mucosa, but can extend into the submucosa (Figure 8). They are mainly composed of small to medium-sized centrocytes next to a few interspersed centroblasts. One characteristic feature is infiltration of the tumor cells into the villi giving them a “glove balloon sign” [85]. The immunophenotype is identical to that of nodal cFL with reactivity for CD10, BCL6 and BCL2, and there is also frequent expression of IgA and alpha4 beta7 which is a mucosal homing receptor [86]. The meshwork of follicular dendritic cells has been described as distorted and pushed to the periphery of the neoplastic follicles (so-called “hollow” meshworks) [87]. The proliferation index is typically low.

DTFL harbor the t(14;18)/*IGH::BCL2*-fusion and mutations typically encountered in cFL such as in *TNFRSF14*, *EZH2*, *KMT2D* and *CREBBP* [84]. The frequency of *KMT2D* mutations is lower than in sFL counterparts, as is genetic complexity [23]. The analysis of the GE profile revealed some differences of DTFL in comparison with cFL [88]. CCL20 is strongly expressed in DTFL which recruits pro-inflammatory Th17+. Usage of IGH variable regions is skewed towards VH4 and VH5 suggesting an antigen driven process and a role for chronic inflammation [89].

## 10. Primary Cutaneous Follicle Center Lymphoma

Im WHO-HAEM5, primary cutaneous follicle center lymphoma (PCFCL) has been upgraded to a definite entity and conceptually set apart from the FL family due to its distinctive clinical features. PCFCL is confined to the skin and usually manifests in the deep dermis [90,91]. Owing to its usually indolent clinical course, no grading had been recommended already in WHO-HAEM4R. On histology, PCFCL can have a purely follicular, mixed follicular and diffuse or entirely diffuse growth pattern and is mainly composed of centrocytes with varying proportions of centroblasts (Figure 9). Immunohistochemistry shows reactivity for the GC-associated antigens CD10 and BCL6, however, BCL2 expression is usually not seen or weak [92,93,94]. In keeping with this, typical PCFCL does not harbor the *IGH::BCL2* fusion, however, rare cases with exclusive manifestation in the skin have been described that show the t(14;18), and these may have a higher risk for subsequent systemic disease [94,95]. The tumor cells have the immunoglobulin genes clonally rearranged and show somatic hypermutation of IGHV genes [96,97,98]. Mutations in genes typically affected in nodal FL are seen in much lower frequency in PCFCL, such as *BCL2* (0%), *CREBBP* and *KTM2D* (20–25%) [99]. N-linked glycosylation motives have been identified in the B-cell receptor genes, obviously providing a proliferative signal via interaction with lectins [100].

The prognosis is usually excellent, and cutaneous relapses seen in around 30% of patients do not negatively affect outcome.

## 11. Perspective

From the molecular point of view, impressive progress has been achieved within the past decade, teaching a lot about the genetic landscape of different FL subtypes, although future in-depth investigations are still needed. Of pivotal impact, our increasing knowledge contributes to our ability of relating distinctive histomorphologic features with the underlying biology that varies between the different FL entities. This is also of relevance in the clinical management of patients with FL, who might benefit from a more intricate molecular classification for optimized risk stratification. The complexity of the various tumor phenotypes and the essential crosstalk of the tumor and the non-malignant bystander cells need to be further characterized, in particular in the context of clinical trials, to better understand the molecular mechanisms mediating a response to different immunomodulatory drugs, antibodies and chemotherapeutic agents. In order to improve preclinical drug testing in FL novel, innovative models have been developed that are highly promising in future perspectives [61,101].

## 12. Conclusions and Future Directions

There has been a vast increase in genetic and molecular data regarding follicular lymphoma within the last decade. The classification of FL in WHO-HAEM5 provides a robust framework for the diagnosis of these neoplasms incorporating new findings and adjusting them to the task of pathologists in providing valid diagnoses. At the same time, this classification and the more sophisticated deciphering of the molecular basis of their origin and progression provides a solid template for future work, ultimately for the sake of patients.

## Figures and Tables

**Figure 1 cancers-15-00785-f001:**
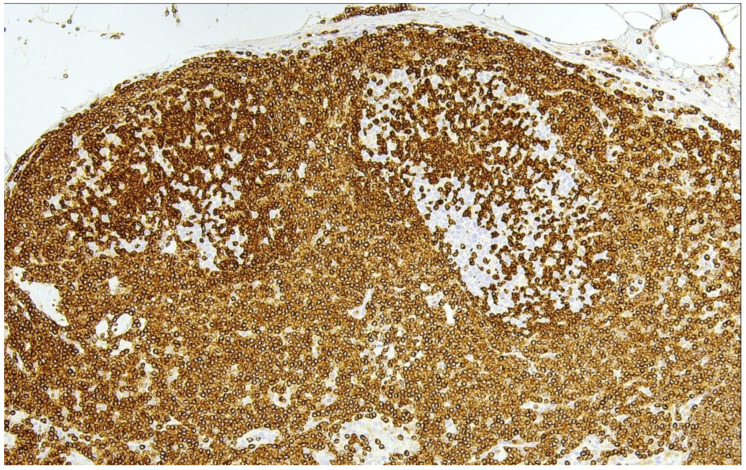
In-situ follicular B-cell neoplasm (H&E ×100). This photomicrograph illustrates strong aberrant reactivity for BCL2 in two germinal centers showing a slightly stronger staining than the surrounding marginal zone B- and T-cells. The overall nodal architecture is well preserved.

**Figure 2 cancers-15-00785-f002:**
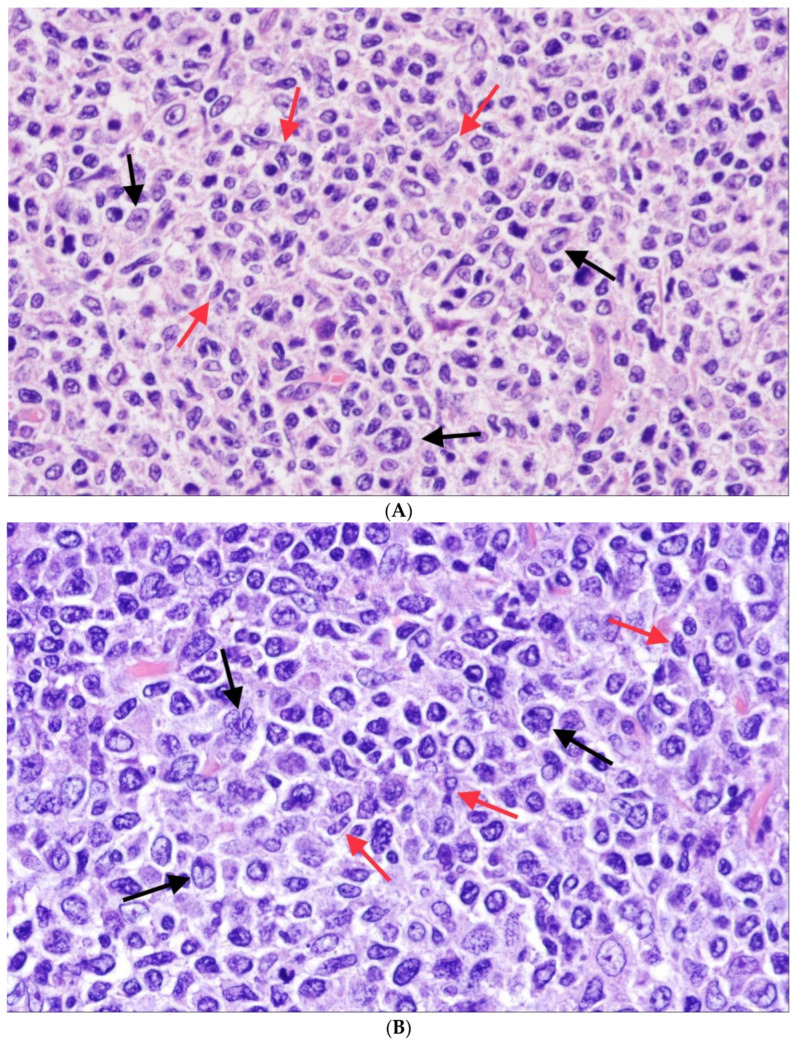
Classic follicular lymphoma (cFL) (H&E ×400). (**A**) This is a typical example of classic FL (FL grade 1/2 according to WHO-HAEM4R) with a predominance of centrocytes (red arrows) and only a few interspersed centroblasts (black arrows). (**B**) In this example, the number of large, transformed cells (centroblasts, black arrows) is distinctly higher (>15 centroblasts per high-power-field (FL grade 3A according to WHO-HEAM4R), however, centrocytes (red arrows) are still present.

**Figure 3 cancers-15-00785-f003:**
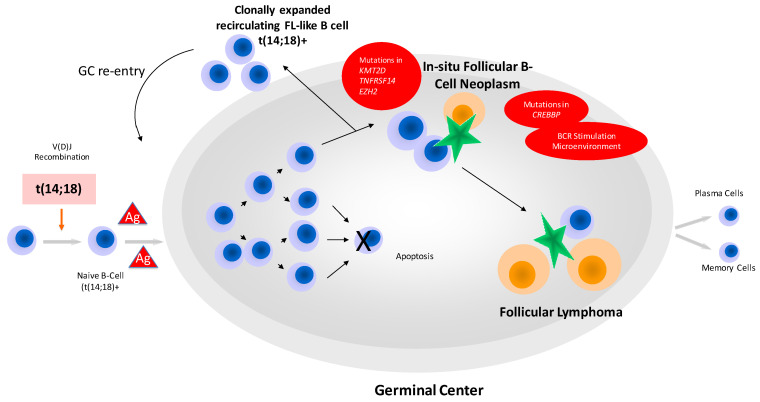
A schematic overview on the evolution of follicular lymphoma. The t(14;18) arose in pre-B- cells during failures in VDJ joining and leads to the generation of long-lived FL-like B-cells. Via several re-entries into germinal centers, these t(14;18)-positive B-cells acquire additional mutations enabling them to be founders of ISFN or manifest lymphoma. Orange cells and green stars represent T-cells and follicular dendritic cells, respectively. Under physiological conditions germinal center B-cells undergo apoptosis, while this process is inhibited in FL (marked by X).

**Figure 4 cancers-15-00785-f004:**
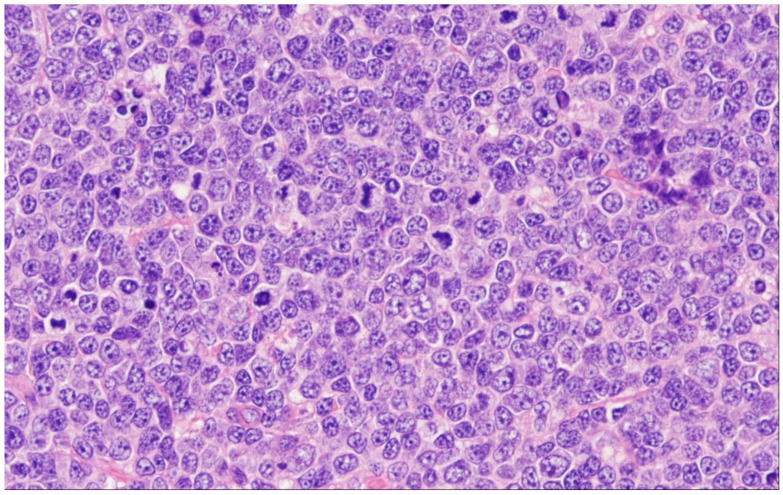
Follicular lymphoma with unusual cytological features (H&E ×400). In this example of a neoplasm growing in follicular structures, there is a predominance of small to medium-sized blastoid cells with round nuclei, finely dispersed chromatin and in part small nucleoli. Note the increased number of mitotic figures. This case had a proliferation index of 80%.

**Figure 5 cancers-15-00785-f005:**
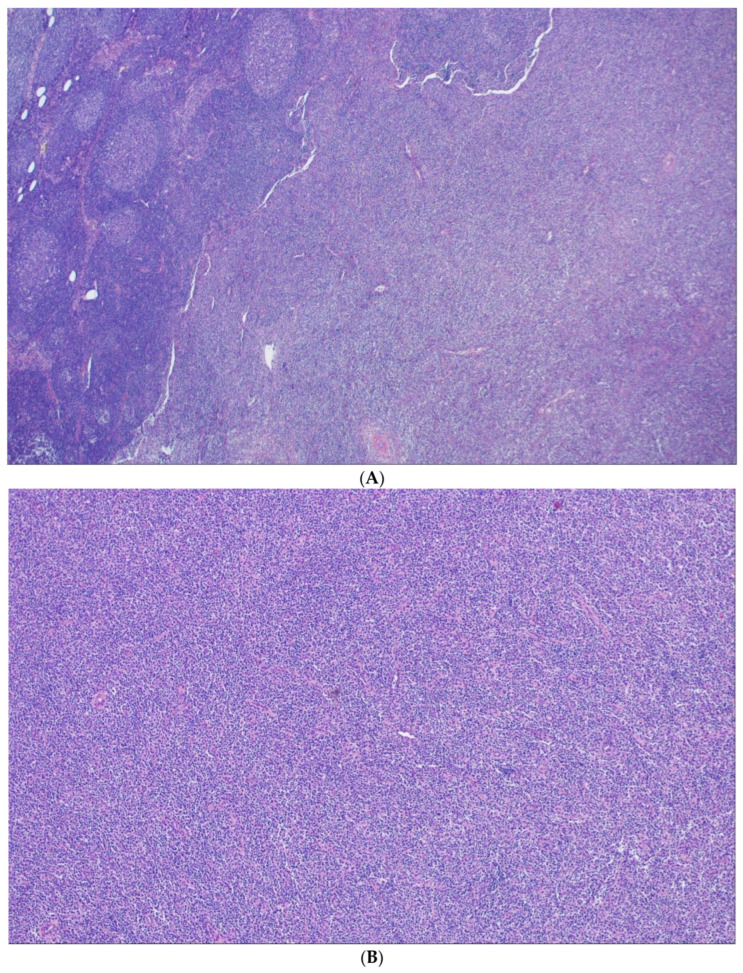
Predominantly diffuse FL. (**A**) Low magnification shows a preserved subcapsular rim of preserved reactive lymphatic tissue to the left and an ill-defined lymphomatous infiltration to the right (H&E ×40). (**B**) Higher magnification reveals an entirely diffuse growth pattern (H&E ×100).

**Figure 6 cancers-15-00785-f006:**
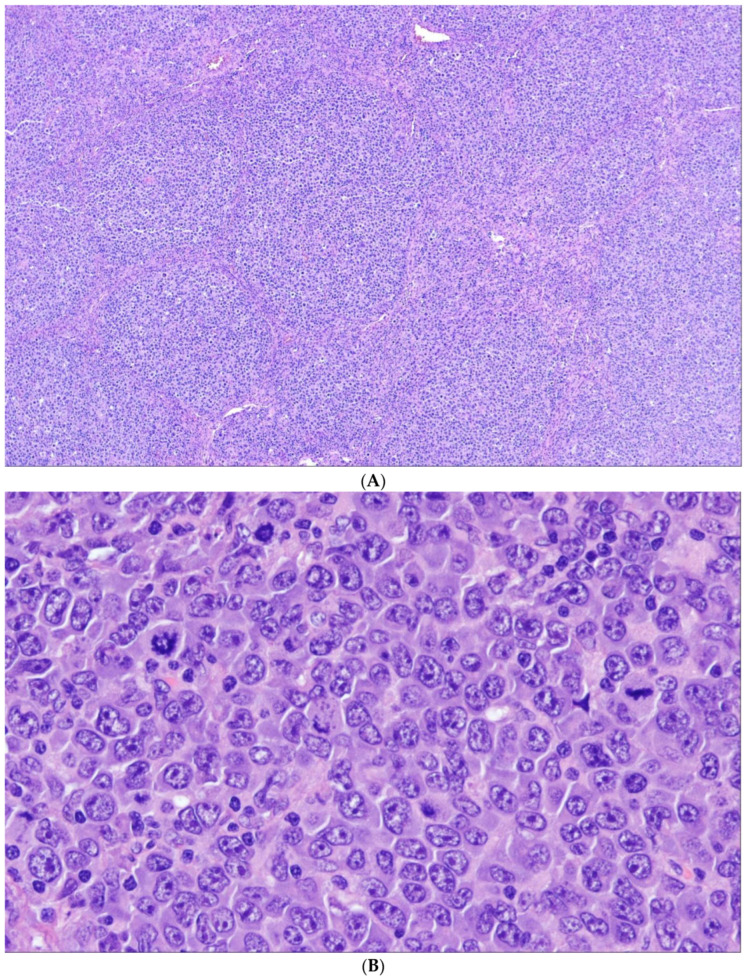
Follicular large B-cell lymphoma. (**A**) There is a clearly discernible follicular growth pattern (H&E ×40). (**B**) On high magnification, the neoplastic follicles are composed of large, transformed cells (centroblasts) exclusively, without an admixture of (typical) centrocytes (H&E ×400).

**Figure 7 cancers-15-00785-f007:**
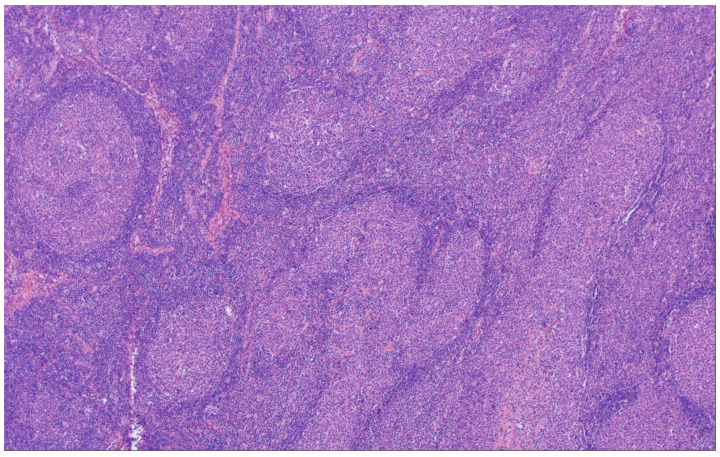
Pediatric-type follicular lymphoma (PTFL) (H&E ×100). In PTFL, there is an intriguing proliferation of large follicles, often lacking mantle zones, and that are large and expansile, often with a “serpiginous” growth pattern.

**Figure 8 cancers-15-00785-f008:**
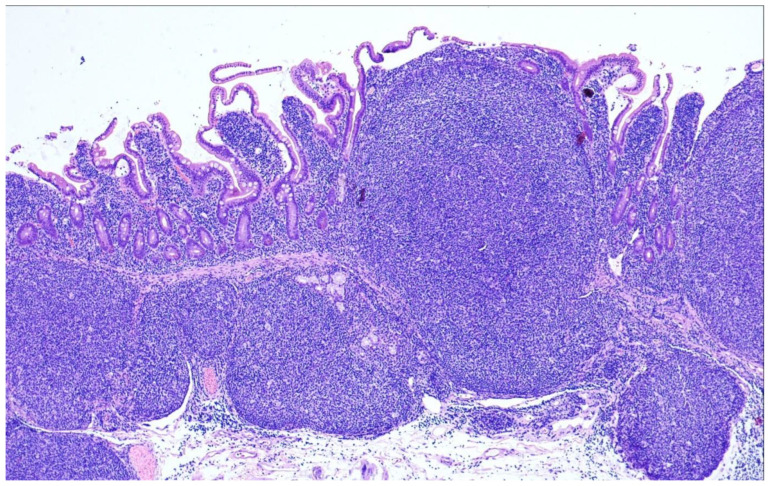
Duodenal-type follicular lymphoma (H&E ×40). Low magnification shows large, atypical nodules/follicles occupying the mucosa and extending into the upper part of the submucosa in this duodenal resection specimen. There is an absence of polarization of the follicles that are composed nearly exclusively of small centrocytes.

**Figure 9 cancers-15-00785-f009:**
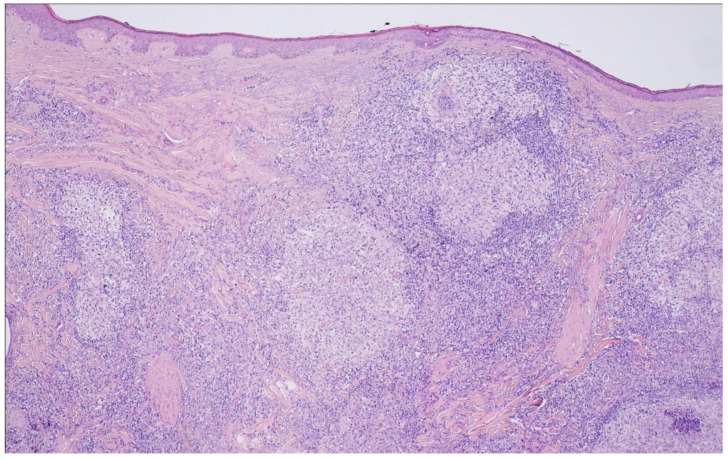
Primary cutaneous follicle center lymphoma (H&E ×20). This example shows atypical follicular structures occupying the dermal portions of the skin underneath an intact epidermis.

**Table 1 cancers-15-00785-t001:** Overview of FL subtypes as described in the WHO-HAEM5.

Follicular Lymphoma (FL)
- Classic FL
- Predominantly diffuse FL
- FL with unusual cytological features
- Follicular large B-cell lymphoma
In situ follicular B-cell neoplasm
Paediatric-type follicular lymphoma
Duodenal-type follicular lymphoma
Primary cutaneous follicle center lymphoma

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
