# Peer review of "Follicular Lymphoma in the 5th Edition of the WHO-Classification of Haematolymphoid Neoplasms—Updated Classification and New Biological Data"

_cancers, 2023, doi:10.3390/cancers15030785_

Round 1

Reviewer 1 Report

The article provides a comprehensive review and classification updates for follicular lymphoma and its various subtypes. The articles is well-written with excellent illustrations, figures, and references. 

I have no edits but I would strongly recommend adding a small paragraph comparing the WHO 5th edition classification to the new ICC classification. Although my own personal view is that we should stick with a unified WHO classification and not revert back to the old days of multiple classifications, the emergence of such classification should be acknowledged in the submitted paper in terms of similarities/ differences.

Author Response

Point-by-point response to the Reviewers:

Reviewer 1
The article provides a comprehensive review and classification updates for follicular lymphoma and its various subtypes. The articles is well-written with excellent illustrations, figures, and references.

I have no edits but I would strongly recommend adding a small paragraph comparing the WHO 5th edition classification to the new ICC classification. Although my own personal view is that we should stick with a unified WHO classification and not revert back to the old days of multiple classifications, the emergence of such classification should be acknowledged in the submitted paper in terms of similarities/ differences.

In accordance to the reviewer’s comment we now added comparative information to the ICC classification where necessary.

Reviewer 2 Report

1- English: The manuscript could benefit from editing for grammar, missing words, and subject-verb agreement, etc. It is recommended that authors delete irrelevant "general" phrases and sentences, repeated and unneeded words. They should use short sentences. Also, some Introductory sentences are irrelevant or are not needed. There are also typos in the manuscript.

2-      Abbreviations: All abbreviations should be revised and defined at their first use. For example, “WHO” should be defined in the abstract.

3-      In scientific writing, in general, symbols for genes are italicized whereas symbols for proteins are not italicized. The formatting of symbols for RNA and complementary DNA (cDNA) usually follows the same conventions as those for gene symbols. Gene names that are written out in full are not italicized (e.g., insulin-like growth factor 1). Genotype designations should be italicized, whereas phenotype designations should not be italicized. 

4-      References: some need to be updated.

5-      Abstract: “large B-cell lymphoma (FLBCL),. In contrast” remove the comma.

6-      Abstract: “cytology more frequently display” should be “cytology more frequently displays.”

7-      Introduction: Very well written.

8-      In the new WHO classification, in-situ follicular B-cell neoplasm, paediatric-type follicular lymphoma, and duodenal-type follicular lymphoma are considered as distinct entities separate from FL. This should be made clear in the manuscript.

9-      Add objective or magnification used in Figure 1. Also, is this image adopted from the WHO online blue books? Same applies to the rest of the figures.

10-  It is recommended that authors add arrows to point out centrocytes and centroblasts.

Author Response

Point-by-point response to the Reviewers:

Reviewer 2

English: The manuscript could benefit from editing for grammar, missing words, and subject-verb agreement, etc. It is recommended that authors delete irrelevant "general" phrases and sentences, repeated and unneeded words. They should use short sentences. Also, some Introductory sentences are irrelevant or are not needed. There are also typos in the manuscript.

We thank the reviewer for this comment and now revised the manuscript accordingly

2-      Abbreviations: All abbreviations should be revised and defined at their first use. For example, “WHO” should be defined in the abstract.

We now revised all abbreviations, also in the abstract.

3-      In scientific writing, in general, symbols for genes are italicized whereas symbols for proteins are not italicized. The formatting of symbols for RNA and complementary DNA (cDNA) usually follows the same conventions as those for gene symbols. Gene names that are written out in full are not italicized (e.g., insulin-like growth factor 1). Genotype designations should be italicized, whereas phenotype designations should not be italicized.

We thank the reviewer for this comment and now revised the manuscript accordingly

4-      References: some need to be updated.

Done

5-      Abstract: “large B-cell lymphoma (FLBCL),. In contrast” remove the comma.

Done

6-      Abstract: “cytology more frequently display” should be “cytology more frequently displays.”

Done

7-      Introduction: Very well written.

8-      In the new WHO classification, in-situ follicular B-cell neoplasm, paediatric-type follicular lymphoma, and duodenal-type follicular lymphoma are considered as distinct entities separate    from FL. This should be made clear in the manuscript.

We thank the reviewer for alerting us on this point and now changed accordingly.

9-      Add objective or magnification used in Figure 1. Also, is this image adopted from the WHO online blue books? Same applies to the rest of the figures.

We thank the reviewer for this comment and now added this missing information to the figure legends. All figures have been taken/constructed for this paper. They are not adopted from the WHO blue book.

10-  It is recommended that authors add arrows to point out centrocytes and centroblasts.

Done